# Unified representation of tractography and diffusion-weighted MRI data using sparse multidimensional arrays

**Cesar F. Caiafa**[*]
Department of Psychological and Brain Sciences
Indiana University (47405) Bloomington, IN, USA
IAR - CCT La Plata, CONICET / CIC-PBA
(1894) V. Elisa, ARGENTINA
ccaiafa@gmail.com

**Olaf Sporns**
Department of Psychological and Brain Sciences
Indiana University (47405) Bloomington, IN, USA
osporns@indiana.edu

**Andrew J. Saykin**
Department of Radiology - Indiana University
School of Medicine. (46202) Indianapolis, IN, USA
asaykin@iupui.edu

**Franco Pestilli**[†]
Department of Psychological and Brain Sciences
Indiana University (47405) Bloomington, IN, USA
franpest@indiana.edu

## Abstract

Recently, linear formulations and convex optimization methods have been proposed to predict diffusion-weighted Magnetic Resonance Imaging (dMRI) data given estimates of brain connections generated using tractography algorithms. The size of the linear models comprising such methods grows with both dMRI data and connectome resolution, and can become very large when applied to modern data. In this paper, we introduce a method to encode dMRI signals and large connectomes, i.e., those that range from hundreds of thousands to millions of fascicles (bundles of neuronal axons), by using a sparse tensor decomposition. We show that this tensor decomposition accurately approximates the Linear Fascicle Evaluation (LiFE) model, one of the recently developed linear models. We provide a theoretical analysis of the accuracy of the sparse decomposed model, LiFE$_{\text{SD}}$, and demonstrate that it can reduce the size of the model significantly. Also, we develop algorithms to implement the optimization solver using the tensor representation in an efficient way.

---
[*]http://web.fi.uba.ar/~ccaiafa/Cesar.html
[†]http://www.brain-life.org/plab/

# 1   Introduction

Multidimensional arrays, hereafter referred to as *tensors*, are useful mathematical objects to model a variety of problems in machine learning [2, 47] and neuroscience [27, 8, 50, 48, 3, 26, 13]. Tensor decomposition algorithms have a long history of applications in signal processing, however, only recently their relation to sparse representations has started to be explored [35, 11]. In this work, we present a sparse tensor decomposition model and its associated algorithm applied to diffusion-weighted Magnetic Resonance Imaging (dMRI).

Diffusion-weighted MRI allows us to estimate structural brain connections in-vivo by measuring the diffusion of water molecules at different spatial directions. Brain connections are comprised of a set of *fascicles* describing the putative position and orientation of the neuronal axons bundles wrapped by myelin sheaths traveling within the living human brain [25]. The process by which fascicles (*the connectome*) are identified from dMRI measurements is called *tractography*. Tractography and dMRI are the primary methods for mapping structural brain networks and white matter tissue properties in living human brains [6, 46, 34]. Despite current limits and criticisms, through these methods we have learned much about the macrostructural organization of the human brain, such that network neuroscience has become one of the fastest-growing scientific fields [38, 43, 44].

In recent years, a large variety of tractography algorithms have been proposed and tested on modern datasets such as the Human Connectome Project (HCP) [45]. However, it has been established that the estimated anatomical properties of the fascicles depend on data type, tractography algorithm and parameters settings [32, 39, 7]. Such variability in estimates makes it difficult to trust a single algorithm for all applications, and calls for routine statistical evaluation methods of brain connectomes [32]. For this reason, linear methods based on convex optimization have been proposed for connectome evaluation [32, 39] and simultaneous connectome and white matter microstructure estimation [15]. However, these methods can require substantial computational resources (memory and computation load) making it prohibitive to apply them to the highest resolution datasets.

In this article, we propose a method to encode brain connectomes in multidimensional arrays and perform statistical evaluation efficiently on high-resolution datasets. The article is organized as follows: in section 2, the connectome encoding method is introduced; in section 2.1, a linear formulation of the connectome evaluation problem is described; in section 3, the approximated tensor decomposed model is introduced; in section 3.3, we derive a theoretical bound of the approximation error and compute the theoretical compression factor obtained with the tensor decomposition; in section 4 we develop algorithms to make the operations needed for solving the connectome evaluation optimization problem; in section 5 we present experimental results using high resolution *in vivo* datasets; finally, in section 6, the main conclusions of our work are outlined.

# 2   Encoding brain connectomes into multidimensional array structures.

We propose a framework to encode brain connectome data (both dMRI and white matter fascicles) into tensors [12, 11, 23] to allow fast and efficient mathematical operations on the structure of the connectome. Here, we introduce the tensor encoding framework and show how it can be used to implement recent methods for statistical evaluation of tractography [32]. More specifically, we demonstrate that the framework can be used to approximate the Linear Fascicle Evaluation model [32] with high accuracy while reducing the size of the model substantially (with measured compression factors up to 40x). Hereafter, we refer to the new tensor encoding method as ENCODE [10]. ENCODE maps fascicles from their natural brain space (Fig. 1(a)) into a three dimensional sparse tensor $\underline{\mathbf{\Phi}}$ (Fig. 1(b)). The first dimension of $\underline{\mathbf{\Phi}}$ (1st mode) encodes each individual white matter fascicle's orientation at each position along their path through the brain. Individual segments (nodes) in a fascicle are coded as non-zero entries in the sparse array (dark-blue cubes in Fig. 1(b)). The second dimension of $\underline{\mathbf{\Phi}}$ (2nd mode) encodes each fascicle's spatial position within dMRI data volume (voxels). Slices in this second dimension represent single voxels (cyan lateral slice in Fig. 1(b)). The

third dimension (3$^{\text{rd}}$ mode) encodes the indices of each fascicle within the connectome. Full fascicles are encoded as $\underline{\mathbf{\Phi}}$ frontal slices (c.f., yellow and blue in Fig. 1(b)).

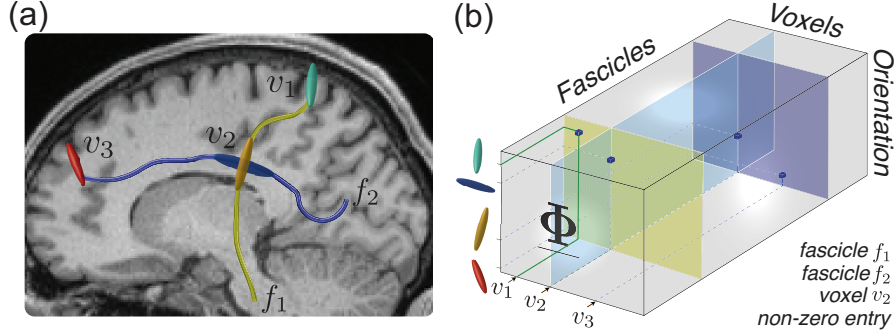

Figure 1: The ENCODE method: mapping structural connectomes from natural brain space to tensor space. (a) Two example white matter fascicles ($f_1$ and $f_2$) passing through three voxels ($v_1$, $v_2$ and $v_3$). (b) Encoding of the two fascicles in a three dimensional tensor. The non-zero entries in $\underline{\mathbf{\Phi}}$ indicate fascicle's orientation (1$^{\text{st}}$ mode), position (voxel, 2$^{\text{nd}}$ mode) and identity (3$^{\text{rd}}$ mode).

Below we demonstrate how to use ENCODE to integrate connectome each fascicle's structure and measured dMRI signal into a single tensor decomposition model. We then show how to use this decompositon model to implement very efficiently a recent model for tractography evaluation, the linear fascicle evaluation method, also referred to as LiFE [32]. Before introducing the tensor decomposition method, we briefly describe the LiFE model, as this is needed to explain the model decomposition using the ENCODE method. We then calculate the theoretical bounds to accuracy and compression factor that can be achieved using ENCODE and tensor decomposition. Finally, we report the results of experiments on real data and validate the theoretical calculations.

## 2.1 Statistical evaluation for brain connectomes by convex optimization.

The Linear Fascicle Evaluation (LiFE) method was introduced to compute the statistical error of the fascicles comprising a structural brain connectome in predicting the measured diffusion signal [32]. The fundamental idea behind LiFE is that a connectome should contain fascicles whose trajectories represent the measured diffusion signal well. LiFE implements a method for connectome evaluation that can be used, among other things, to eliminate tracked fascicles that do not predict well the diffusion signal. LiFE takes as input the set of fascicles generated by using tractography methods (the candidate connectome) and returns as output the subset of fascicles that best predict the measured dMRI signal (the optimized connectome). Fascicles are scored with respect to how well their trajectories represent the measured diffusion signal in the voxels along the their path. To do so, weights are assigned to each fascicle using convex optimization. Fascicles assigned a weight of zero are removed from the connectome, as their contribution to predicting the diffusion signal is null. The following linear system describes the equation of LiFE (see Fig. 2(a)):

$$\mathbf{y} \approx \mathbf{M}\mathbf{w}, \tag{2.1}$$

where $\mathbf{y} \in \mathbb{R}^{N_{\boldsymbol{\theta}} N_v}$ is a vector containing the demeaned signal $y_i = \bar{S}(\boldsymbol{\theta}_{n_i}, v_i)$ measured at all white-matter voxels $v_i \in \mathcal{V} = \{1, 2, \ldots, N_v\}$ and across all diffusion directions $\boldsymbol{\theta}_n \in \Theta = \{\boldsymbol{\theta}_1, \boldsymbol{\theta}_2, \ldots, \boldsymbol{\theta}_{N_{\boldsymbol{\theta}}}\} \subset \mathbb{R}^3$, and $\mathbf{w} \in \mathbb{R}^{N_f}$ contains the weights for each fascicle in the connectome.

Matrix $\mathbf{M} \in \mathbb{R}^{N_{\boldsymbol{\theta}} N_v \times N_f}$ contains, at column $f$, the predicted demeaned signal contributed by fascicle $f$ at all voxels $\mathcal{V}$ and across all directions $\Theta$:

$$\mathbf{M}(i, f) = S_0(v_i) O_f(\boldsymbol{\theta}_{n_i}, \mathbf{v}_f). \tag{2.2}$$

$S_0(v)$ is defined as the *non diffusion-weighted signal* and $O_f(\boldsymbol{\theta}, \mathbf{v}_f)$ is the *orientation distribution function* [32] of fascicle $f$ at diffusion direction $\boldsymbol{\theta}$, i.e.

$$O_f(\boldsymbol{\theta}, \mathbf{v}_f) = e^{-b(\boldsymbol{\theta}^T \mathbf{v}_f)^2} - \frac{1}{N_{\boldsymbol{\theta}}} \sum_{\boldsymbol{\theta}_n \in \Theta} e^{-b(\boldsymbol{\theta}_n^T \mathbf{v}_f)^2}, \tag{2.3}$$

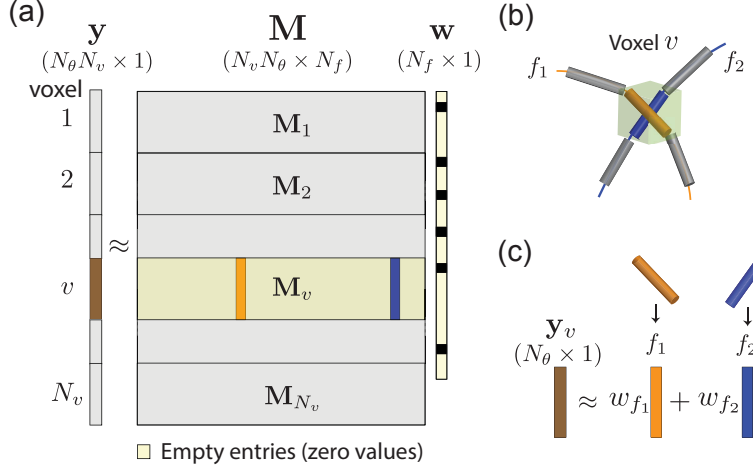

Figure 2: The Linear Fascicle Evaluation (LiFE) model. (a) The predicted signal $\mathbf{y} \in \mathbb{R}^{N_\theta N_v}$ in all voxels and gradient directions is obtained by multiplying matrix $\mathbf{M} \in \mathbb{R}^{N_\theta N_v \times N_f}$ by the vector of weights $\mathbf{w} \in \mathbb{R}^{N_f}$ (see equation 2.1). (b) A voxel containing two fascicles, $f_1$ and $f_2$. (c) The predicted diffusion signal $\mathbf{y}_v \in \mathbb{R}^{N_\theta}$ at voxel $v$ is approximated by a nonnegative weighted linear combination of the predicted signals for the fascicles in the voxel.

where the simple "stick" diffusion tensor model [31] was used and vector $\mathbf{v}_f \in \mathbb{R}^3$ is defined as the spatial orientation of the fascicle in that voxel.

Whereas vector $\mathbf{y}$ and matrix $\mathbf{M}$ in equation (2.1) are fully determined by the dMRI measurements and the output of a tractography algorithm, respectively, the vector of weights $\mathbf{w}$ needs to be estimated by solving a Non-Negative Least squares (NNLS) optimization problem, which is defined as follows:

$$\min_{\mathbf{w}} \left( \frac{1}{2} \|\mathbf{y} - \mathbf{M}\mathbf{w}\|^2 \right) \text{ subject to } w_f \geq 0, \forall f. \tag{2.4}$$

As a result, a sparse non-negative vector of weights $\mathbf{w}$ is obtained. Whereas nonzero weights correspond to fascicles that contribute to predict the measured dMRI signal, fascicles with zero weight make no contribution to predicting the measurements and can be eliminated. In this way, LiFE identifies the fascicles supported by the data in a candidate connectome providing a principled approach to evaluate connectomes in terms of prediction error as well as the number of non-zero weighted fascicles.

A noticeable property of the LiFE method is that the size of matrix $\mathbf{M}$ in equation (2.1) can require tens of gigabytes for full-brain connectomes, even when using optimized sparse matrix formats [19]. Below we show how to use ENCODE to implement a sparse tensor decomposition [9, 11] of matrix $\mathbf{M}$. This decomposition allows accurate approximation of the original LiFE model with dramatic reduction in memory requirements.

## 3 Theoretical results: Tensor decomposition and approximation of the linear model for tractography evaluation.

We describe the theoretical approach to factorizing the LiFE model, eq. (2.1). We note that matrix $\mathbf{M} \in \mathbb{R}^{N_\theta N_v \times N_f}$ (Fig. 2(a)) can be rewritten as a tensor (3D-array) $\underline{\mathbf{M}} \in \mathbb{R}^{N_\theta \times N_v \times N_f}$ by decoupling the gradient direction and voxel indices into separate indices, i.e. $\underline{\mathbf{M}}(n_i, v_i, f) = \mathbf{M}(i, f)$, where $n_i = \{1, 2, \ldots, N_\theta\}$, $v_i = \{1, 2, \ldots, N_v\}$ and $f = \{1, 2, \ldots, N_f\}$. Thus, equation (2.1) can be rewritten in tensor form as follows:

$$\mathbf{Y} \approx \underline{\mathbf{M}} \times_3 \mathbf{w}^T, \tag{3.1}$$

where $\mathbf{Y} \in \mathbb{R}^{N_\theta \times N_v}$ is obtained by converting vector $\mathbf{y} \in \mathbb{R}^{N_\theta N_v}$ into a matrix (matricization) and "$\times_n$" is the tensor-by-matrix product in mode-$n$ [23], more specifically, the mode-3

product in the above equation is defined as follows: $\mathbf{Y}(n, v) = \sum_{f=1}^{N_f} \underline{\mathbf{M}}(n, v, f)\mathbf{w}_f$. Below, we show how to approximate the tensor model in equation (3.1) using a sparse Tucker decomposition [9] by first focusing on the dMRI signal in individual voxels and then across voxels.

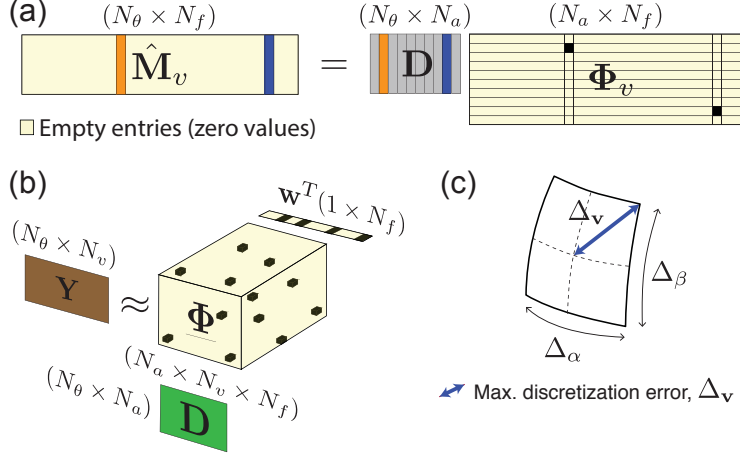

Figure 3: The LiFE$_{\text{SD}}$ model: (a) Each block $\mathbf{M}_v$ of matrix $\mathbf{M}$ (a lateral slice in tensor $\underline{\mathbf{M}}$) is factorized by using a dictionary of diffusion signal predictions $\mathbf{D}$ and a sparse matrix of coefficients $\mathbf{\Phi}_v$. (b) LiFE$_{\text{SD}}$ model is written as a Tucker decomposition model with a sparse core tensor $\underline{\mathbf{\Phi}}$ and factors $\mathbf{D}$ (mode-1) and $\mathbf{w}^T$ (mode-3). (c). The maximum distance between a fascicle orientation vector $\mathbf{v}$ and its approximation $\mathbf{v}_a$ is determined by the discretization of azimuth ($\Delta_\alpha$) and elevation ($\Delta_\beta$) spherical coordinates. More specifically, for $\Delta_\alpha = \Delta_\beta = \pi/L$, the maximum discretization error is $\|\mathbf{\Delta}_{\mathbf{v}}\| \leq \frac{\pi}{\sqrt{2}L}$.

## 3.1 Approximation of the linear model within individual brain voxels.

We focus on writing the linear formulation of the diffusion prediction model (Fig. 2(b)-(c)) by restricting equation (3.1) to individual voxels, $v$:

$$\mathbf{y}_v \approx \mathbf{M}_v\mathbf{w}, \tag{3.2}$$

where vector $\mathbf{y}_v = \mathbf{Y}(:, v) \in \mathbb{R}^{N_\theta}$ and matrix $\mathbf{M}_v = \underline{\mathbf{M}}(:, v, :) \in \mathbb{R}^{N_\theta \times N_f}$, correspond to a column in $\mathbf{Y}$ and a lateral slice in tensor $\underline{\mathbf{M}}$, respectively. We propose to factorize matrix $\mathbf{M}_v$ as follows

$$\mathbf{M}_v \approx \hat{\mathbf{M}}_v = \mathbf{D}\mathbf{\Phi}_v, \tag{3.3}$$

where matrix $\mathbf{D} \in \mathbb{R}^{N_\theta \times N_a}$ is a dictionary of diffusion predictions whose columns (atoms) correspond to precomputed fascicle orientations, and $\mathbf{\Phi}_v \in \mathbb{R}^{N_a \times N_f}$ is a sparse matrix whose non-zero entries, $\mathbf{\Phi}_v(a, f)$, indicate the orientation of fascicle $f$ in voxel $v$, which is approximated by atom $a$ (see Fig. 3(a) for an example of a voxel $v$ as shown in Fig. 2(b)-(c)). For computing the diffusion predictions, we use a discrete grid in the sphere by uniformly sampling the spherical coordinates using $L$ points in azimuth and elevation coordinates (Fig. 2(c)).

## 3.2 Approximation of the linear model across multiple brain voxels.

By applying the approximation introduced in equation (3.3) to every slice in tensor $\underline{\mathbf{M}}$ in equation 3.1, we obtain the following tensor Sparse Decomposed LiFE model, hereafter referred to as LiFE$_{\text{SD}}$ (Fig. 3(b)):

$$\mathbf{Y} \approx \underline{\mathbf{\Phi}} \times_1 \mathbf{D} \times_3 \mathbf{w}^T, \tag{3.4}$$

where $\mathbf{D}$ is a common factor in mode-1, i.e., it multiplies all lateral slices. It is noted that, the formula in the above equation (3.4), is a particular case of the Tucker decomposition [42, 16] where the core tensor $\underline{\mathbf{\Phi}}$ is sparse [9, 11], and only factors in mode-1 ($\mathbf{D}$) and mode-3 ($\mathbf{w}^T$)

are present. By comparing equations (3.4) and (3.1) we define the LiFE$_\text{SD}$ approximated tensor model as

$$\hat{\underline{\mathbf{M}}} = \underline{\mathbf{\Phi}} \times_1 \mathbf{D} \tag{3.5}$$

## 3.3 Theoretical bound for model decomposition accuracy and data compression.

In this section, we derive a theoretical bound on the accuracy of LiFE$_\text{SD}$ compared to the original LiFE model (Proposition 3.1) and we theoretically analyze the compression factor associated to the factorized tensor approximation (Proposition 3.2). Hereafter, we assume that, in a given connectome having $N_f$ fascicles, each fascicle has a fixed number of nodes $(N_n)$, and the diffusion weighted measurements were taken on $N_{\boldsymbol{\theta}}$ gradient directions with a gradient strength $b$. The proofs of the propositions can be found in the Supplementary material.

**Proposition 3.1** (accuracy). For a given connectome, and dictionary $\mathbf{D}$ obtained by uniformly sampling the azimuth-elevation $(\alpha, \beta)$ space using $\Delta_\alpha = \Delta_\beta = \pi/L$ (see Fig. 3(c)), the following upper bound on the Frobenius norm based model error is verified:

$$\|\underline{\mathbf{M}} - \hat{\underline{\mathbf{M}}}\|_F \leq \frac{2b\pi\sqrt{6N_f N_n N_{\boldsymbol{\theta}}}}{L}. \tag{3.6}$$

The importance of this theoretical result is that the error is inversely proportional to the discretization parameter $L$, which allows one to design the decomposed model so that a prescribed accuracy is met.

**Proposition 3.2** (size reduction). For a given connectome, and a dictionary $\mathbf{D} \in \mathbb{R}^{N_{\boldsymbol{\theta}} \times N_a}$ containing $N_a$ atoms (columns of matrix $\mathbf{D}$), the achieved compression factor is

$$CF = \left( \frac{4}{3N_{\boldsymbol{\theta}}} - \frac{N_a}{3N_n N_f} \right)^{-1}, \tag{3.7}$$

where $CF = C(\underline{\mathbf{M}})/C(\hat{\underline{\mathbf{M}}})$, with $C(\underline{\mathbf{M}})$ and $C(\hat{\underline{\mathbf{M}}})$ being the storage costs of LiFE and LiFE$_\text{SD}$ models, respectively.

It is noted that, usually $3N_n N_f \gg N_a$, which implies that the compression factor can be approximated by $CF \approx \frac{3N_{\boldsymbol{\theta}}}{4}$, i.e., it is proportional to the number of gradient directions $N_{\boldsymbol{\theta}}$.

# 4 Model optimization using tensor encoding.

Once the LiFE$_\text{SD}$ model has been built, the final step to validate a connectome requires finding the non-negative weights that least-squares fit the measured diffusion data. This is a convex optimization problem that can be solved using a variety of NNLS optimization algorithms. We used a NNLS algorithm based on first-order methods specially designed for large scale problems [22]. Next, we show how to exploit the decomposed LiFE$_\text{SD}$ model in the optimization.

The gradient of the original objective function for the LiFE model can be written as follows:

$$\nabla_{\mathbf{w}} \left( \frac{1}{2} \|\mathbf{y} - \mathbf{Mw}\|^2 \right) = \mathbf{M}^T \mathbf{Mw} - 2\mathbf{M}^T \mathbf{y}, \tag{4.1}$$

where $\mathbf{M} \in \mathbb{R}^{N_{\boldsymbol{\theta}} N_v \times N_f}$ is the original LiFE model, $\mathbf{w} \in \mathbb{R}^{N_f}$ the fascicle weights and $\mathbf{y} \in \mathbb{R}^{N_{\boldsymbol{\theta}} N_v}$ the demeaned diffusion signal. Because the decomposed version does not explicitly store $\mathbf{M}$, below we describe how to perform two basic operations ($\mathbf{y} = \mathbf{Mw}$ and $\mathbf{w} = \mathbf{M}^T \mathbf{y}$) using the sparse decomposition.

## 4.1 Computing $\mathbf{y} = \mathbf{Mw}$

Using equation (3.1) we can see that the product $\mathbf{Mw}$ can be computed using equation (3.4) and vectorizing the result, i.e. $\mathbf{y} = vec(\mathbf{Y})$, where $vec()$ stands for the vectorization

operation, i.e., to convert a matrix to a vector by stacking its columns in a long vector. In Algorithm 1, we present the steps for computing $\mathbf{y} = \mathbf{M}\mathbf{w}$ in an efficient way.

---

**Algorithm 1 : y = M_times_w($\underline{\Phi}$,D,w)**

---

**Require:** Decomposition components ($\underline{\Phi}$, $\mathbf{D}$ and vector $\mathbf{w} \in \mathbb{R}^{N_f}$).
**Ensure: y = Mw**
 1: $\mathbf{Y} = \underline{\Phi} \times_3 \mathbf{w}^T$; `the result is a large but very sparse matrix` $(N_a \times N_v)$
 2: $\mathbf{Y} = \mathbf{D}\mathbf{Y}$; `the result is a relatively small matrix` $(N_\theta \times N_v)$
 3: $\mathbf{y} = vec(\mathbf{Y})$
 4: **return y;**

---

## 4.2 Computing $\mathbf{w} = \mathbf{M}^T\mathbf{y}$

The product $\mathbf{w} = \mathbf{M}^T\mathbf{y}$ can be computed using LiFE$_{\text{SD}}$ in the following way:

$$\mathbf{w} = \mathbf{M}^T\mathbf{y} = \mathbf{M}_{(3)}\mathbf{y} = \mathbf{\Phi}_{(3)}(\mathbf{I} \otimes \mathbf{D}^T)\mathbf{y}, \qquad (4.2)$$

where $\mathbf{M}_{(3)} \in \mathbb{R}^{N_f \times N_\theta N_v}$ and $\mathbf{\Phi}_{(3)} \in \mathbb{R}^{N_f \times N_a N_v}$ are the *unfolding matrices* [23] of tensors $\underline{\mathbf{M}} \in \mathbb{R}^{N_\theta \times N_v \times N_f}$ and $\underline{\mathbf{\Phi}} \in \mathbb{R}^{N_a \times N_v \times N_f}$, respectively; $\otimes$ is the Kronecker product and $\mathbf{I}$ is the $(N_v \times N_v)$ identity matrix. Equation (4.2) can be written also as follows [9]:

$$\mathbf{w} = \mathbf{\Phi}_{(3)}vec(\mathbf{D}^T\mathbf{Y}). \qquad (4.3)$$

Because matrix $\mathbf{\Phi}_{(3)}$ is very sparse, we avoid computing the large and dense matrix $\mathbf{D}^T\mathbf{Y}$ and compute instead only its blocks that are being multiplied by the non-zero entries in $\mathbf{\Phi}_{(3)}$. This allows maintaining efficient memory usage and limits the number of CPU cycles needed. In Algorithm 2, we present the steps for computing $\mathbf{w} = \mathbf{M}^T\mathbf{y}$ in an efficient way.

---

**Algorithm 2 : w = Mtransp_times_y($\underline{\Phi}$,D,y)**

---

**Require:** Decomposition components ($\underline{\mathbf{\Phi}}$, $\mathbf{D}$) and vector $\mathbf{y} \in \mathbb{R}^{N_\theta N_v}$.
**Ensure: $\mathbf{w} = \mathbf{M}^T\mathbf{y}$**
 1: $\mathbf{Y} \in \mathbb{R}^{N_\theta \times N_v} \leftarrow \mathbf{y} \in \mathbb{R}^{N_\theta N_v}$; `reshape vector y into a matrix Y`
 2: $[\mathbf{a}, \mathbf{v}, \mathbf{f}, \mathbf{c}] = $ get_nonzero_entries($\underline{\mathbf{\Phi}}$); $a(n)$, $v(n)$, $f(n)$, $c(n)$ indicate the atom, the voxel, the fascicle and the entry in tensor $\underline{\mathbf{\Phi}}$ associated to node $n$, respectively, with $n = 1, 2, \ldots, N_n$;
 3: $\mathbf{w} = \mathbf{0} \in \mathbb{R}^{N_f}$; `Initialize weights with zeros`
 4: **for** $n = 1$ **to** $N_n$ **do**
 5:     $w(f(n)) = w(f(n)) + \mathbf{D}^T(:, a(n))\mathbf{Y}(:, v(n))c(n)$;
 6: **end for**
 7: **return w;**

---

# 5 Experimental results: Validation of the theoretical bounds for model decomposition accuracy and data compression.

Here, we validate our theoretical findings by using dMRI data from subjects in a public source (the Stanford dataset [32]). The data were collected using $N_{\boldsymbol{\theta}} = 96$ (STN96, five subjects) and $N_{\boldsymbol{\theta}} = 150$ (STN150, one subject) directions with b-value $b = 2,000 s/mm^2$. We performed tractography using these data and both, probabilistic and deterministic methods, in combination with Constrained Spherical Deconvolution (CSD) and the diffusion tensor model (DTI) [41, 17, 5]. We generated candidate connectomes with $N_f = 500,000$ fascicles per brain brain. See for [10, 32, 39] for additional details on data preprocessing.

We first analyzed the accuracy of the approximated model (LiFE$_{\text{SD}}$) as a function of the parameter, $L$, which describes the number of fascicles orientations encoded in the dictionary $\mathbf{D}$. In theory, the larger the number of atoms in $\mathbf{D}$ the higher the accuracy of the approximation. We show that model error (defined as $e_{\mathbf{M}} = \frac{\|\mathbf{M} - \hat{\mathbf{M}}\|_F}{\|\underline{\mathbf{M}}\|_F}$) decreases as a function of the parameter $L$ for all subjects in the dataset Fig. 4(a). This result validates the theoretical upper bound in Proposition 3.1. We also solved the convex optimization problem of equation

(2.4) for both, LiFE and LiFE$_{\text{SD}}$, and estimated the error in the weights assigned to each fascicle by the two models (we computed the error in weights as follows $e_{\mathbf{w}} = \frac{\|\mathbf{w}-\hat{\mathbf{w}}\|}{\|\mathbf{w}\|}$). Fig. 4(b) shows the error $e_{\mathbf{w}}$ as a function of the parameter $L$. It is noted that for $L > 180$ the error is lower than 0.1% in all subjects.

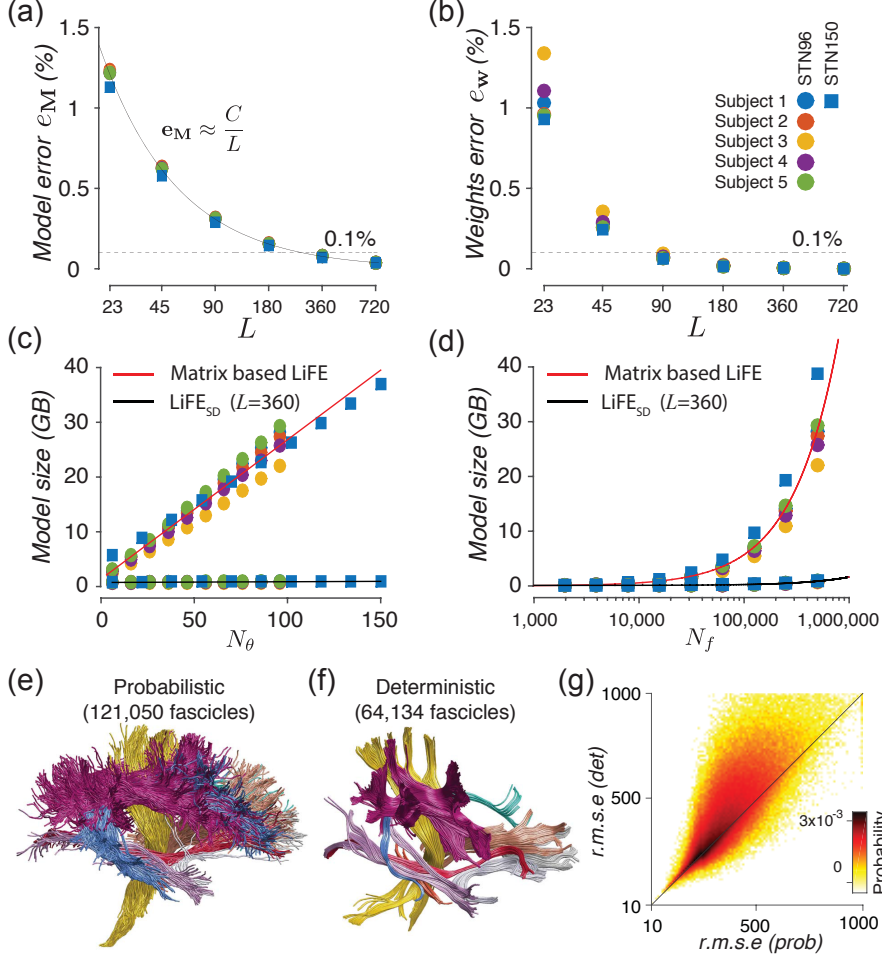

Figure 4: Experimental results: (a) The model error $e_{\mathbf{M}}$ in approximating the matrix $\mathbf{M}$ with LiFE$_{\text{SD}}$ is inversely proportional to the parameter $L$ as predicted by our Proposition 3.1 ($\mathbf{e_M} \approx C/L$ was fitted to the data with $C = 27.78$ and a fitting error equal to 2.94%). (b) Error in the weights obtained by LiFE$_{\text{SD}}$ compared with original LiFE's weights, $e_{\mathbf{w}}$, as a function of parameter $L$. (c)-(d) Model size (GB) scales linearly with the number of directions $N_\theta$ and the number of fascicles $N_f$, however it increases much faster in the LiFE model compared to the LiFE$_{\text{SD}}$ model. LiFE$_{\text{SD}}$ was computed using $L = 360$. (e)-(f) Probabilistic and deterministic connectomes validated with LiFE$_{\text{SD}}$ for a HCP subject. (g) Comparison of the Root-mean-squared-error (r.m.s, as defined in [32]) obtained in all voxels for probabilistic and deterministic connectomes. The averaged r.m.s.e are 361.12 and 423.06 for the probabilistic and deterministic cases, respectively.

Having experimentally demonstrated that model approximation error decreases as function of $L$, we move on to demonstrate the magnitude of model compression achieved by the tensor decomposition approach. To do so, we fixed $L = 360$ and computed the model size for both, LiFE and LiFE$_{\text{SD}}$, as a function of the number of gradient directions $N_{\boldsymbol{\theta}}$ (Fig. 4(c)) and fascicles $N_f$ (Fig. 4(d)). Results show that, as predicted by our theoretical results in Proposition 3.2, model size scales linearly with the number of directions for both, LiFE and LiFE$_{\text{SD}}$, but that the difference in slope is profound. Experimentally measured compression ratios raise up to approximately 40 as it is the case for the subjects in the STN150 dataset ($N_f = 500,000$ and $N_{\boldsymbol{\theta}} = 150$).

Finally, we show an example comparison between two connectomes obtained by applying probabilistic [17] and deterministic [4] tracking algorithms to one brain dataset (a single subject) from the Human Connectome Project dataset [45], with $N_{\boldsymbol{\theta}} = 90$, $N_v = 267,306$ and $N_f = 500,000$. Figs. 4e-f show the detected 20 major tracts in a human brain using only the fascicles with nonzero weigths. In this case, the probabilistic connectome has more fascicles $(121,050)$ than the deterministic one $(64,134)$. Moreover, we replicate previous results demonstrating that probabilistic connectomes have lower error than the deterministic one in a majority of the voxels (see Fig. 4(g)).

## 6   Conclusions

We introduced a method to encode brain connectomes in multidimensional arrays and decomposition approach that can accurately approximate the linear model for connectome evaluation used in the LiFE method [32]. We demonstrate that the decomposition approach dramatically reduces the memory requirements of the LiFE model, approximately from 40GB to 1GB, with a small model approximation error of less than 1%. The compactness of the decomposed LIFE model has important implications for other computational problems. For example, model optimization can be implemented by using operations involving tensorial operations avoiding the use of large matrices such as $\mathbf{M}$ and using instead the sparse tensor and prediction dictionary ($\underline{\boldsymbol{\Phi}}$ and $\mathbf{D}$ respectively).

Multidimensional tensors and decomposition methods have been used to help investigators make sense of large multimodal datasets [27, 11]. Yet to date these methods have found only a few applications in neuroscience, such as performing multi-subject, clustering and electroencephalography analyses [49, 48, 3, 28, 26, 13, 8]. Generally, decomposition methods have been used to find compact representations of complex data by estimating the combination of a limited number of common meaningful factors that best fit the data [24, 27, 23]. We propose a new application that, instead of using the decomposition to estimate latent factors, it encodes the structure of the problem explicitly.

The new application of tensor decomposition proposed here has the potential to improve future generations of models of connectomics, tractography evaluation and microstructure [32, 15, 36, 39]. Improving these models will allow going beyond the current limitations of the state of the art methods [14]. Finally, tensorial representations for brain imaging data have the potential to contribute advancing the application of machine learning algorithms to mapping the human connectome [18, 37, 21, 20, 30, 1, 51, 29, 40, 33].

**Acknowledgments**

This research was supported by (NSF IIS-1636893; BCS-1734853; NIH ULTTR001108) to F.P. Data provided by Stanford University (NSF BCS 1228397). F.P. were partially supported by the Indiana University Areas of Emergent Research initiative Learning: Brains, Machines, Children.

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
