[Supplementary Material]

# Unified representation of tractography and diffusion-weighted MRI data using sparse multidimensional arrays
# SUPPLEMENTARY MATERIAL

**Cesar F. Caiafa**[*]
Department of Psychological and Brain Sciences
Indiana University (47405) Bloomington, IN, USA
IAR - CCT La Plata, CONICET / CIC-PBA
(1894) V. Elisa, ARGENTINA
ccaiafa@gmail.com

**Olaf Sporns**
Department of Psychological and Brain Sciences
Indiana University (47405) Bloomington, IN, USA
osporns@indiana.edu

**Andrew J. Saykin**
Department of Radiology - Indiana University
School of Medicine. (46202) Indianapolis, IN, USA
asaykin@iupui.edu

**Franco Pestilli**[†]
Department of Psychological and Brain Sciences
Indiana University (47405) Bloomington, IN, USA
franpest@indiana.edu

## 1   Proofs of the theoretical bound for accuracy and compression factor

In this section, we derive a theoretical bound on the accuracy of LiFE$_{\text{SD}}$ compared to the original LiFE model (Proposition 3.1) and we theoretically analyze the compression factor associated to the factorized tensor approximation (Proposition 3.2). Hereafter, we assume a given connectome having $N_f$ fascicles, each fascicle having a fixed number of $N_n$ nodes, and where diffusion weighted measurements were taken on $N_{\boldsymbol{\theta}}$ gradient directions with a gradient strength $b$.

*Proof of Proposition 3.1:* The error in modeling the diffusion signal for a particular voxel $v$, fascicle $f$ and gradient direction $\boldsymbol{\theta}$ is given by:

$$\Delta_{\mathbf{O}}(\boldsymbol{\theta}) = |\mathbf{O}_f(\boldsymbol{\theta}) - \mathbf{D}(\boldsymbol{\theta}, a)|, \tag{S1}$$

where $\mathbf{O}_f(\boldsymbol{\theta})$ is the orientation distribution function as defined in equation (2.3) (we avoided making reference to the voxel $v$ for clearity) and $\mathbf{D}(\boldsymbol{\theta}, a)$ is the diffusion signal of atom $a$

---

[*]http://web.fi.uba.ar/~ccaiafa/Cesar.html
[†]http://www.brain-life.org/plab/

at gradient direction $\boldsymbol{\theta} = [\theta_x, \theta_y, \theta_z]^T$. By defining $\mathbf{v} = [v_x, v_y, v_z]^T$ and $\mathbf{v}_a = \mathbf{v} + \boldsymbol{\Delta}_\mathbf{v} = [v_x + \Delta_{v_x}, v_y + \Delta_{v_y}, v_z + \Delta_{v_z}]^T$ as the vectors pointing out at the directions of the fascicle $f$ and its closest dictionary atom $a$, respectively (see Fig. 3c), we arrive at:

$$\Delta_\mathbf{O}(\boldsymbol{\theta}) = |\Delta_g(\boldsymbol{\theta}) - \frac{1}{N_\theta} \sum_{\theta'} \Delta_g(\boldsymbol{\theta}')|, \tag{S2}$$

where $\Delta_g = |g(\mathbf{v}_1 + \boldsymbol{\Delta}_\mathbf{v}, \boldsymbol{\theta}) - g(\mathbf{v}, \boldsymbol{\theta})|$ with $g(\mathbf{v}, \boldsymbol{\theta}) = e^{-b(\boldsymbol{\theta}^T\mathbf{v})^2}$. For a sufficiently small error vector $\boldsymbol{\Delta}_\mathbf{v} = [\Delta_{v_x}, \Delta_{v_y}, \Delta_{v_z}]^T$, we can approximate $\Delta_g(\boldsymbol{\theta})$ as follows:

$$\Delta_g(\boldsymbol{\theta}) \approx \left| \frac{\partial g(\mathbf{v}, \boldsymbol{\theta})}{\partial v_x} \right| \Delta_{v_x} + \left| \frac{\partial g(\mathbf{v}, \boldsymbol{\theta})}{\partial v_y} \right| \Delta_{v_y} + \left| \frac{\partial g(\mathbf{v}, \boldsymbol{\theta})}{\partial v_z} \right| \Delta_{v_z}, \tag{S3}$$

and, by using the fact that $|\boldsymbol{\theta}^T\mathbf{v}| \leq 1$, $e^{-b(\boldsymbol{\theta}^T\mathbf{v})^2} \leq 1$, $\Delta_{v_x}, \Delta_{v_y}, \boldsymbol{\Delta}_{\mathbf{v}_z} \leq \|\boldsymbol{\Delta}_\mathbf{v}\| \leq \frac{\pi}{\sqrt{2}L}$, and $\|\boldsymbol{\theta}\|_1 \leq \sqrt{3}\|\boldsymbol{\theta}\|$ in equation (S3), we obtain: $\Delta_g(\mathbf{v}, \boldsymbol{\theta}) \leq \frac{b\pi\sqrt{6}}{L}$. Thus, by using this result in equation (S2), we obtain an upper bound for the error of modeling the diffusion signal of one fascicle and at one gradient direction in a voxel: $\Delta_\mathbf{O}(\boldsymbol{\theta}) \leq \frac{2b\pi\sqrt{6}}{L}$. Finally, by summing up all over the nodes in the connectome, it implies that

$$\|\underline{\mathbf{M}} - \hat{\underline{\mathbf{M}}}\|_F^2 \leq N_f N_n N_{\boldsymbol{\theta}} \left( \frac{2b\pi\sqrt{6}}{L} \right)^2. \tag{S4}$$

$\square$

*Proof of Proposition 3.2:* The memory load necessary to store each fascicle in a sparse matrix $\mathbf{M}$ is $3N_{\boldsymbol{\theta}}N_n$, because using a sparse matrix structure, three numbers are required for each node, i.e., the row-column indices plus the entry value. Thus the storage cost of $\mathbf{M}$ is:

$$C(\mathbf{M}) = \mathcal{O}(3N_n N_{\boldsymbol{\theta}} N_f). \tag{S5}$$

Conversely, storing fascicles in the LiFE$_{\text{SD}}$ model requires $4N_n$ values per fascicle plus the dictionary matrix (i.e., the set of the non-zero entries and their locations within the tensor $\underline{\boldsymbol{\Phi}}$ plus matrix $\mathbf{D}$). Thus, the amount of memory required in the LiFE$_{\text{SD}}$ model is:

$$C(\hat{\underline{\mathbf{M}}}) = \mathcal{O}(4N_n N_f + N_{\boldsymbol{\theta}} N_a), \tag{S6}$$

where $N_{\boldsymbol{\theta}} N_a$ is the storage associated with the dictionary matrix $\mathbf{D} \in \mathbb{R}^{N_{\boldsymbol{\theta}} \times N_a}$. Finally, by taking the ratio of equations (S5) and (S6), we arrive at the expression of the compression factor as shown in equation (3.7). $\square$