[Reviews · NeurIPS 2017]

Reviewer 1



The paper propose an efficient approximation of the "Linear Fascicle Evaluation model" (LiFE), which is used for tractography. The key issue with the LiFE model is that it computationally scales poorly such that it is impractical to analyze MRI data at full resolution. The authors propose a sparse approximation based on a Tucker model (tensor model). The authors prove that the approximation is both good and achieves a high compression; this is further evaluated experimentally. The model appears solid and likewise for the analysis. My key objection to the paper is that the actual evaluation of the model for tractography is treated very superficially (just a couple of lines of text). I understand that within the tight boundaries of a NIPS paper, the authors does not have much space for such an evaluation, so I'm willing to forgive this limitation of the paper. Other than that, the paper is very well written with intuitive figures guiding the reader along the way. As a non-expert, I found the paper a please to read. I do stress that I'm not an expert in neither tractography or tensor models. A few minor comments (not important in the rebuttal): *) In line 36, the authors point out that tractography is poorly understood and that competing methods can give wildly different results. The authors then argue that this motivates convex models. I don't buy this argument -- the reason different models give different results is more likely due to different modeling assumptions, rather than whether a global optimum has been found or not. *) After the propositions: the authors take the time to explain in plain words the key aspects of their propositions. Most authors neglect this, and I just want to say that such efforts make the paper much easier to read. So: thank you! *) In line 195, you write: "...dataset Fig. 4.a." There's something wrong with that sentence. Perhaps the figure reference should be in a parenthesis? *) In line 202: typo: "aproach" --> "approach". == Post rebuttal == I have read the other reviews and the rebuttal, and stand by my evaluation that this paper should be accepted.

Reviewer 2



The authors present a memory-efficient, approximate implementation of the LiFE method for track filtering/reweighing in diffusion MRI. The LiFE method takes any input set of white matter fibre tracks and assigns each of them a weight such that their overall distribution matches the apparent fibre density in the diffusion MRI data. In the original work, this was described as a large non-negative least squares (NNLS) problem. In this work, the design matrix is approximated using discrete atoms associated with fibre orientation, enabling to cast the NNLS problem as a memory-efficient, sparse tensor representation. Performance of the approximation, as well as memory efficiency of the tensor encoding, are demonstrated in human brain data. The paper is well-written and clear. The conclusions are supported by the data. The work is an important contribution, of interest to the diffusion MRI tractography community. I have only 2 points of criticism: 1. I disagree with the terminology "decomposition" and "factorization" in the title and throughout the text. This vocabulary is commonly associated with methods such as the singular value decomposition and nonnegative matrix factorization, in which a single matrix or tensor is represented as a linear combination of _unknown_ matrices or tensors, computed _simultaneously_ under certain constraints. In this work, matrix M and its sparse tensor representation D and Phi are precomputed as inputs to the method. At its core, LiFE(sd) is computing the remaining nonnegative weights, only one factor in the tensor representation. 2. The stick model only accounts for restricted, intra-axonal diffusion, ignoring hindered extra-cellular diffusion or free water compartments. Furthermore, the axial diffusivity seems to be fixed at 1 ms/um^2 in this work. How appropriate is this model and its fixed parameter value for these data? At what b-value was the data acquired and is this sufficiently high to assume that CSF and extra-cellular diffusion are sufficiently attenuated?

Reviewer 3



Summary: This paper extends the matrix-based LiFE model, which can be used to represent the measured diffusion signal (dRMI) by the trajectories of fascicles contained in brain connectomes generated via tractography algorithm, to a tensor-based LiFE_sd model by factorizing the 3rd-order encoded candidate connectome tensor into coefficient core tensor Phi along with factor dictionary D and weight vector w. In this way, the LiFE_sd with compact multi-linear representation can significantly reduce the size of LiFE model while maintaining the approximation accuracy. Strength: 1. The paper applies tensor decomposition methods on a novel application of estimating the brain contectome to improve the potentiality on the big data, which could be significantly important in neuroscience. 2. Some analysis results such as accuracy and size compression are availbe when comparing with matrix-based model. 3. The paper is clearly written. Weakness: 1. From the methodology aspect, the novelty of paper appears to be rather limited. The ENCODE part is already proposed in [10] and the incremental contribution lies in the decomposition part which just factorizes the M_v into factor D and slices Phi_v. 2. For the experiment, I'd like to the effect of optimized connectome in comparison with that of LiFE model, so we can see the performance differences and the effectiveness of the tensor-based LiFE_sd model. This part of experiment is missing.